# Spongy moths from Europe and Asia: Who could have higher invasion risk in North American?

Yi Luo[1,2‡], Changxi Li[1,2‡], Xiaokang Hu[1,2]*, Jianmeng Feng[1,2]*

1 College of Agriculture and Biological Science, Dali University, Dali, Yunnan, China, 2 Cangshan Forest Ecosystem Observation and Research Station of Yunnan Province, Dali University, Dali, China

‡ Co-first author
* dluhxk@yeah.net (XH); fengdlu@163.com (JF)

## Abstract

North American forest systems are significantly impacted by spongy moths (*Lymantria dispar* Linnaeus). It is unclear, nevertheless, how are the invasion risks of spongy moths from Asia and Europe in North American relative to each other. In this study, we compared the potential ranges of spongy moths from Asia (ASM) and those from Europe (ESM) in North America, and investigated the range shifts between spongy moths in North America (NASM) and ASM and ESM. ASM and ESM would occupy larger potential ranges in North America than NASM, i.e., 7.16 and 6.98 times, respectively. Thus, one should not undervalue the invasive potential posed by spongy moths from Asia and Europe. Compared to ESM, ASM displayed larger ranges in North America. It is likely due to ASM's tolerance of more variable climates. Consequently, even though ASM was more recently introduced to North America than ESM, it's possible that the former has higher invasion risk in North American.

## 1. Introduction

Originating from Eurasia, the spongy moth (*Lymantria dispar* Linnaeus) stands as one of the most damaging invasive pests globally. Since its introduction to North America over a century ago, this species has proliferated, spreading to 20 American states and 5 provinces in Canada [1]. Its presence has led to severe defoliation, and the death of trees or shrubs, contributing to the disruption of both native and commercial forest ecosystems, whether through direct or indirect means [2–5]. Notably, it has defoliated more than 98 million acres of forests since 1924, resulting in substantial economic and ecological detriments [6]. The expenses associated with managing this invasive pest in North America have been estimated at $3.2 billion annually [7]. Despite considerable investments and the implementation of initiatives such as the Slow the Spread Program, the spongy moth's proliferation in North America, primarily within the United States, remains unchecked [8]. Therefore, comprehensive investigations into the invasive potential of this pest in North America warrant significant focus.

**Data availability statement:** All relevant data are within the paper and its Supporting Information files.

**Funding:** National Natural Science Foundation of China (Grant ID: 42263012). The funders had no role in study design, data collection and analysis, decision to publish, or preparation of the manuscript.

**Competing interests:** The authors have declared that no competing interests exist.

The spongy moths in Eurasia have two primary native populations: spongy moths in Europe (predominantly consisting of European strains) and spongy moths in Asia (mainly composed of Asian strains), and these populations are native to, and mainly observed in, the continents, from which they derive their names [1,9–12]. Spongy moths from Europe were accidentally introduced to North America in 1869 [13]. Since then, they have inflicted extensive damage upon significant tree species, targeting more than 300 deciduous and coniferous host species [14–17]. Spongy moths from Asia were introduced to North America in 1991 from Russia through the grain trade [18–20], posing new challenges and invasion threats in North America [21]. Their different invasion history in North America may result in their distinct invasion potentials there. Additionally, compared with their European counterparts, spongy moths from Asia have a broader range of hosts [22], encompassing more than 600 plant species; however, not all of these hosts were originally listed for European spongy moths [14,15,23]. Therefore, given the distinct characteristics of spongy moths from Asia and Europe, such as invasion history and hosts [22,24,25], they may possess differing invasive potentials or potential ranges in North America. Nevertheless, their relative potential ranges in North America remain unclear.

The ranges of spongy moths have recruited much attention in recent years [4,5,26–30]. For instance, Régnière et al. examined the potential climatic distribution of spongy moths in Canada, and identified their potential ranges in western Canada [31]. Recently, Nunez-Mir et al. explored vulnerable areas where this invasive pest could potentially spread within the transition zone outlined by the Slow the Spread Program, and they found that seasonal temperatures were crucial in determining the moth's invasive potential [30].

Overall, these studies have focused on the range shifts of this pest driven by future climate changes. Undoubtedly, they have provided valuable information for identifying priority regions, advancing our understanding of the mechanisms behind these pest invasions, and thus shaping our strategies to mitigate their spread. However, studies on the range shifts between the introduced spongy moths in North America and those in native Eurasia are limited. The aforementioned studies offer essential insights for assessing the invasive potentials of this pest in North America.

Climatic conditions are pivotal to the survival of this invasive pest [32–34]. Previous studies have demonstrated that temperature governs enzymatic activities, intestinal microflora, and metabolic rates in spongy moths [35–37]. For example, freezing temperatures can significantly impact the survival and developmental timelines of overwintering eggs [38]. Additionally, compared with those general conditions, microclimate ones could be more direct factors responsible for the life history (e.g., reproduction and behavior) of the spongy moth. For example, the supercooling point of overwintered embryos for this species ranged from -23 to -29℃ [33]. Nevertheless, this species might lay the eggs masses under the snow cover or on the rocks in mountain areas, which could allow its embryos avoid the detrimental effects of lower temperature outside [39,40]. Therefore, microclimate conditions could be the stronger factors limiting the range expansions of this species than general climate

conditions. In sum, climatic conditions could be essential in influencing the spread of this invasive species, and their influences on its potential ranges could not be overlooked [41].

Although climatic conditions significantly impact the potential ranges and range shifts of invasive species, the roles of land use factors should not be underestimated in this context. The reason is that the alterations in land use can modify the habitat characteristics for invasive species, resulting in shifts in their ranges [42,43]. In the present study, land use conditions can, at the very least, influence the availability of suitable hosts. For instance, this invasive pest may encounter fewer hosts in urban areas compared with forested regions. Therefore, considerable attention must be given to understanding the impacts of land use on the range shifts of spongy moths.

The range shifts of this invasive pest are influenced by both land use and climatic factors, but studies on their relative importance, especially concerning spongy moths, are limited [44,45]. Sirami et al. proposed that the influences of climatic conditions relative to those of land use might hinge on parameters such as stronger roles of climatic variables on larger scale while land use on smaller scale [46]. However, the general applicability of this argument requires further investigation. Liu et al. observed weaker effects of climatic variables on the ranges of fall armyworms globally [47]. Conversely, contrasting patterns were observed when examining the ranges of invasive Aedes aegypti on the regional scale [48]. In summary, while both land use and climate changes may contribute to the potential ranges and range shifts of this pest, further studies on their relative effects are urgently needed.

Topographical factors can alter the spatial distribution of water, energy, and solar radiation, resulting in diverse habitats for a wide array of organisms. Moreover, large-scale topographical features, such as deep-cutting valleys and huge mountains, can serve as barriers impeding the dispersal of invasive species [49]. Additionally, although we posited that climatic, topographical, and land use factors could impact the potential ranges and range shifts of this pest, the extent of their individual influences remains uncertain.

In our study, we employed species distribution model tools to examine the potential range of the introduced spongy moths in North America. We also built species distribution models (SDMs) for the native spongy moths from Asia and Europe, separately, and the models were generally determined by their traits or adaptation to the environmental conditions. Then, we transferred the models to North America, and projecting their potential ranges there. We then constructed range shift models to identify the shifts between introduced spongy moths in North America and their native counterparts.

This study categorized spongy moths into three populations: spongy moths from Europe, spongy moths from Asia, and introduced spongy moths in North America which is a combination of European and Asian populations there [11,50]. All potential ranges of spongy moths were projected on the spatial scale of North America.

## 2. Materials and methods

### 2.1. Occurrence records for spongy moths

The principal source of the occurrences for the spongy moths used in this study was the Global Biodiversity Information Facility (https://www.gbif.org, accessed on 4th Jan 2023). It comprises approximately 2 billion occurrence records of species globally, collated from 85,176 datasets, 2031 publishing institutions, and more than 8000 peer-reviewed studies. It is one of the most comprehensive and reliable data sources globally, and has wider acceptance in recent decades.

In addition, we conducted searches for occurrence records from the Commonwealth Agricultural Bureau International (https://www.cabi.org, accessed on 3rd Jan 2023), Early Detection and Distribution Mapping System (https://www.eddmaps.org, accessed on 3rd Jan 2023), and primary scientific literature documenting the occurrences of spongy moths. Our initial occurrence dataset included 35,707 distinct records of spongy moths. We applied spatial distribution model tools to restrict the data to a 5 km radius, as suggested by Yang et al. (2023) [51], to mitigate potential sampling bias effects on our models [52,53]. Hence, our occurrence record dataset comprised 6086 records, consisting of 637 native records from Asia, 3039 native records from Europe, and 2410 introduced records from North America (Fig 1 and S1 Table).

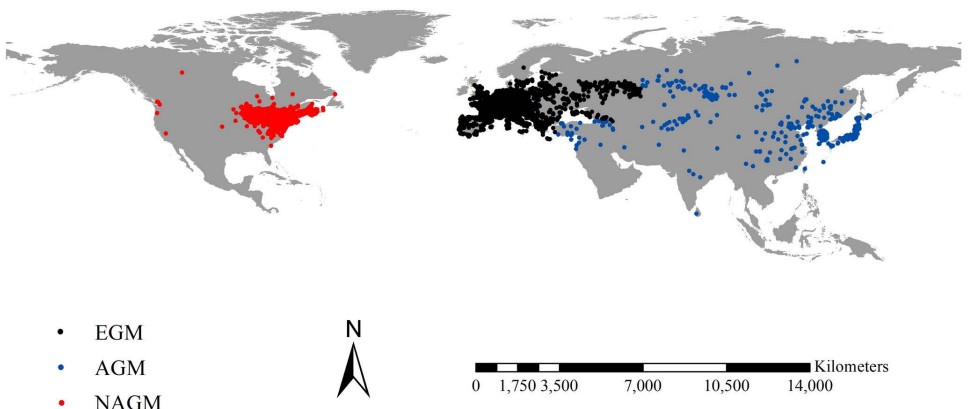

- EGM
- AGM
- NAGM

N

Kilometers
0   1,750 3,500        7,000        10,500        14,000

**Fig 1. Spatial patterns of occurrence records of spongy moths (*L. dispar* Linnaeus).** Red, black, and blue dots indicate spongy moths from North America, Europe, and Asia, respectively. We retrieved 6088 records after the spatial rarefication, including 3676 native records from Eurasia (637 from Asia and 3039 from Europe) and 2410 native records from North America. ESM, ASM, and NASM correspond to spongy moths from Europe, Asia, and North America, respectively. The occurrence records of spongy moths in North America were used to project their potential ranges there. We used the corresponding occurrence records and climate datasets of native spongy moths from Asia and Europe from their native continents to project their potential ranges in North America, individually. Reprinted from Worldclim under a CC BY 4.0 license, with permission from Worldclim, original copyright 2020-2024.

### 2.2. Predictor datasets

Given their influences on the range dynamics of the spongy moth, our 30-predictor datasets included 3 sub-datasets: climate, land use, and topographical datasets (S2 Table). The climate dataset comprised 19 predictors sourced from Worldclim 2.1, widely acknowledged for providing comprehensive climatic data, particularly on a large scale [54]. These 19 climatic predictors with a spatial resolution of 2.5 arcmin encompassed 11 energy availability predictors and 8 climatic predictors, covering annual, quarterly, and monthly climatic variables, as well as seasonality predictors of temperature and precipitation (S2 Table). The land use dataset, derived from the Land-Use Harmonization 2 (LUH2) dataset, featured eight land-use variables with a spatial resolution of 2.5 arc-min (S2 Table). They represented the proportions of various land types, including forested primary land, rangeland, managed pasture, cropland, forested secondary land, non-forested primary land, urban land and non-forested secondary land.

The three topographical predictors, such as slope, aspect, and elevation, were extracted from a digital elevation model which was at a spatial resolution of 0.5 arc-min found in Worldclim 2.0 [55]. Aspect and slope were calibrated using this digital elevation model. To ensure compatibility with the requirements of the SDMs, we processed all predictors to either have a spatial resolution of 2.5 arc-min or be resampled to meet this criterion.

### 2.3. Range projection

**2.3.1. Predictor selection.** We first developed preliminary SDMs to retrieve the relative importance scores for all predictors in the models for introduced spongy moths in North America. We calculated the importance scores for all predictors in the preliminary SDMs through the Jackknife technique (S2 Table). Then, we employed the Pearson correlation coefficient to find strong collinearity among 30 factors (S3 Table). As per Dormann et al.[56], we established a threshold for significant collinearity, considering a Pearson coefficient exceeding 0.7 as indicative of strong collinearity. To reduce multi-collinearity among the predictors, when significant collinearity was detected between pairs of predictors, we systematically removed the predictor with the lower importance value until no further multi-collinearity was detected. The remaining variables then were inputted into the final SDMs to project the range of introduced spongy moths in North America. We employed a similar approach to mitigate multicollinearity among the predictors when modeling the potential ranges of native spongy moths from Asia and Europe separately.

**2.3.2. Range projections for spongy moths.** Biomod2 [57] was employed to predict the ranges of introduced spongy moths in North America, as well as the potential ranges that native spongy moths from Asia and Europe might potentially occupy in North America. To reduce methodological uncertainties in single algorithm, we used seven algorithms in our SDMs: artificial neural network, classification tree analysis, flexible discriminant analysis, maximum entropy model, random forest classifier, generalized additive model, and generalized boosting model. Given the lack of absence data in our models, we generated pseudo-absences (PAs) in SDMs following a widely accepted method [58]. We randomly generated 1000 PAs, or a quantity equivalent to the number of spongy moth occurrences, in case of fewer than 1000 occurrence records. We adopted the sensitivity–specificity sum maximization threshold (MSS threshold), to calibrate the range [59]. We also calibrated the importance scores for the predictors in the final SDMs through the Jackknife technique

**2.3.3. Assessing the model reliability.** Five-time-cross-validation process was conducted to assess SDM performance. During this validation, 70 percentages of the occurrence records were randomly selected to develop SDMs, while the remaining 30 percentages were adopted to gauge the SDMs' reliability [60]. In accordance with the criteria outlined by Fielding and Bell (1997) [61], Allouche et al. (2006) [62], and Gallien et al. (2012) [63], we excluded SDMs with true skill statistics (TSS) below 0.7 and area under the ROC curve (AUC) below 0.8. Furthermore, we followed the approach of Gotelli and Ulrich (2012) [64] and Bohl et al. (2019) to calibrate the reliability of the actual SDMs [65]. Therefore, we constructed null models by randomly generating virtual records in quantities equivalent to 70% of the actual occurrences. These virtual records were then used to develop null SDMs. Subsequently, the remaining 30 percentages of the actual records were employed to evaluate the reliability of the null SDMs. Finally, we conducted the independent-samples $t$ test to compare the AUC and TSS values between the actual and null SDMs [66].

## 2.4. Range shifts in spongy moths

We developed range shift models to investigate the range dynamic between introduced spongy moths in North America and their native counterparts from Asia and Europe. To accomplish this, we categorized the overall ranges of these moths in North America, Asia and Europe into three distinct elements: expansion, stabilization, and unfilling [51]. Expansion signified the range exclusively used by introduced spongy moths in North America, representing their invasiveness. Stabilization referred to the range shared by introduced spongy moths in North America and the native spongy moths from Asia or Europe ($RS$). On the contrary, unfilling represented the range exclusively occupied by native spongy moths in Asia or Europe. The ranges of introduced spongy moths in North America encompassed both the expansion and stabilization elements ($IR$). Conversely, the ranges of native spongy moths in Asia or Europe consisted of a combination of the unfilling and stabilization elements ($NR$). Using range ratio ($RR$), we compare the range between spongy moths in North America and native spongy moths from Asia or Europe ($NR$) to assess the range expansions of the introduced spongy moths in North America relative to those from Asia or Europe [52]. This ratio was expressed as follows:

$$RR = IR/NR$$

When $RR > 1$, the potential ranges of introduced spongy moths were larger than those of the native spongy moths. Additionally, the index of range similarity ($IRS$) was used to calibrate the shifts in range positions between $NR$ and $IR$ [49]. This index could be expressed as follows:

$$IRS = 2RS/(IR + NR)$$

When $IRS > 0.5$, the introduced spongy moths in North America and the native spongy moths from Asia or Europe could be in either similar or different ranges.

## 3. Results

### 3.1. Major predictors in the models

In the models aiming to determine the ranges of introduced spongy moths in North America, the predictors with the highest importance scores were averaged temperature of the warmest season (0.340), followed by annual precipitation (0.190) and temperature seasonality (0.042) (Table 1). In the case of SDMs targeting the potential ranges of native spongy moths from Asia, the dominant predictor was the annual mean temperature (0.185), followed by isothermality (0.074) and precipitation in the wettest month (0.067) (Table 1). In the models of the ranges of native spongy moths from Europe, the most important predictor was the temperature annual range (0.311), followed by the averaged temperature in the warmest season (0.241) and mean diurnal range (0.069) (Table 1). In summary, across all SDMs concerning the potential ranges of spongy moths in from Asia, the climatic factors demonstrated more pronounced influences on their potential ranges compared with land use and topographical factors.

**Table 1. Importance scores of the remaining predictors in final species distribution models for *L. dispar*.**

| *L. dispar* (North America) | | | *L. dispar* (Asia) | | | *L. dispar* (Europe) | | |
|---|---|---|---|---|---|---|---|---|
| Category | Predictor | IS | Category | Predictor | IS | Category | Predictor | IS |
| Clim | Bio10 | 0.34 | Clim | Bio1 | 0.185 | Clim | Bio7 | 0.311 |
| Clim | Bio12 | 0.19 | Clim | Bio3 | 0.074 | Clim | Bio10 | 0.241 |
| Clim | Bio4 | 0.042 | Clim | Bio13 | 0.067 | Clim | Bio2 | 0.069 |
| Topo | Ele | 0.041 | Land | Crop | 0.064 | Clim | Bio3 | 0.067 |
| Land | Range | 0.028 | Topo | Ele | 0.061 | Land | Urban | 0.05 |
| Clim | Bio9 | 0.018 | Land | Urban | 0.056 | Land | Crop | 0.047 |
| Clim | Bio14 | 0.015 | Topo | Slop | 0.038 | Land | Primf | 0.047 |
| Clim | Bio3 | 0.012 | Land | Primf | 0.037 | Land | Pastr | 0.037 |
| Topo | Slop | 0.012 | Land | Range | 0.036 | Clim | Bio9 | 0.031 |
| Clim | Bio18 | 0.006 | Clim | Bio8 | 0.03 | Topo | Slop | 0.015 |
| Clim | Bio8 | 0.005 | Clim | Bio14 | 0.029 | Clim | Bio8 | 0.014 |
| Land | Urban | 0.005 | Land | Secdf | 0.026 | Topo | Ele | 0.013 |
| Land | Crop | 0.004 | Land | Pastr | 0.021 | Land | Secdn | 0.012 |
| Land | Pastr | 0.004 | Clim | Bio2 | 0.015 | Land | Primn | 0.012 |
| Topo | Asp | 0.004 | Land | Primn | 0.013 | Land | Secdf | 0.01 |
| Land | Secdf | 0.003 | Clim | Bio15 | 0.013 | Land | Range | 0.01 |
| Land | Primf | 0.002 | Land | Secdn | 0.012 | Clim | Bio19 | 0.007 |
| Land | Primn | 0.001 | Topo | Asp | 0.009 | Clim | Bio17 | 0.006 |
| Land | Secdn | 0.001 | | | | Clim | Bio18 | 0.004 |
| | | | | | | Clim | Bio15 | 0.003 |
| | | | | | | Topo | Asp | 0.001 |

Clim, Climate; Land, land use; Topo, topography; IS, importance scores; Asp, aspect (°); Crop, cropland; Ele, elevation (m); Primf, forested primary land; Pastr, managed pasture; Range, rangeland; Primn, non-forested primary land; Secdn, non-forested secondary land; Secdf, forested secondary land; Slop, slope (°); Urban, urban land. Bio1, mean annual temperature (℃); Bio2, averaged diurnal ranges (℃); Bio3, isothermality; Bio4, seasonality in temperature; Bio5, max temperature of the warmest months (℃); Bio7, Range of annual temperature (℃); Bio8, averaged temperature of the wettest seasons (℃); Bio9, averaged temperature of the driest seasons (℃); Bio10, averaged temperature of the warmest seasons (℃); Bio12, annual precipitation (mm); Bio13, precipitation of the wettest month (mm); Bio14, precipitation of the driest month (mm); Bio15seasonality in, precipitation (mm); Bio17, precipitation of the driest seasons (mm); Bio18, precipitation of the warmest seasons (mm); Bio19, precipitation of the coldest seasons (mm).

## 3.2. Reliability of the SDMs

The ensemble SDMs calibrating the ranges of introduced spongy moths in North America exhibited a high level of reliability, with the TSS and AUC values reaching 0.938 and 0.995, respectively. They were 0.815 and 0.970 for those from Asia, and 0.699 and 0.930 for those from Europe, respectively. Furthermore, all of the null SDMs exhibited lower reliability compared with their respective real SDMs, with all the *P* values <0.001 (S4 Table).

## 3.3. The ranges of spongy moths

The MSS threshold for calibrating ranges differed across regions: it was 0.60 for introduced spongy moths in North America, and 0.08 and 0.09 for native moths from Asia and Europe, respectively. The range of introduced spongy moths in North America was mostly concentrated in the central eastern region of the United States, specifically in Michigan, Maine, New Brunswick, New Jersey, New York, Virginia, Ohio, Vermont. Pennsylvania, and Wisconsin, covering a substantial area of $112.16 \times 10^4$ km$^2$ (Fig 2a). The range of native spongy moths in Asia were largely identified in southern Canada, encompassing most parts of the United States and nearly the entire territory of Mexico. These ranges covered an extensive area of $1314.84 \times 10^4$ km$^2$. These ranges extended from the east to the west coast, and from Georgia in the south to Alaska in the north (Fig 2b). Similarly, the potential ranges of native spongy moths from Europe exhibited a pattern resembling those of Asian spongy moths. They occupied a significant area of $1074.52 \times 10^4$ km$^2$, extending from the east to the

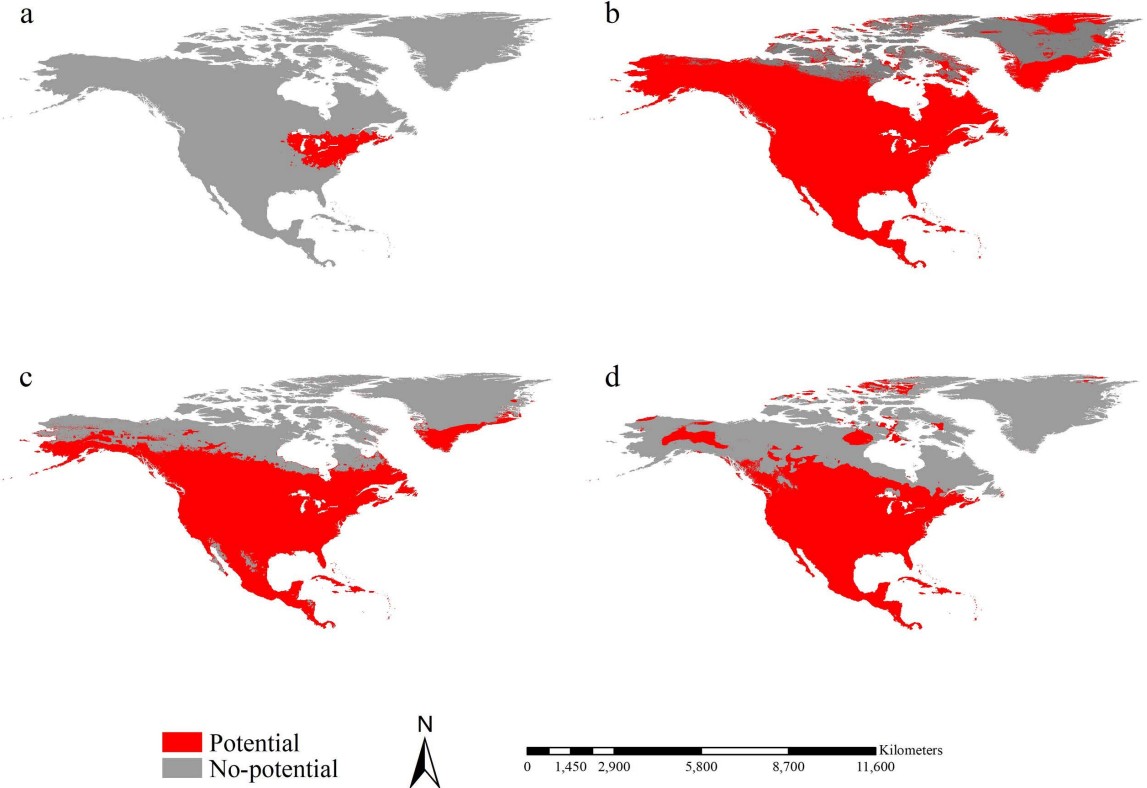

**Fig 2. Potential ranges of spongy moths on the spatial scale of North America.** (a) Potential ranges of introduced spongy moths in North America. (b) Projected potential ranges in North America for the spongy moths from Asia. (c) Projected potential ranges in North America for the spongy moths from Europe. Red and gray colors indicate the potential range and non-potential range in this order. Reprinted from Worldclim under a CC BY 4.0 license, with permission from Worldclim, original copyright 2020-2024.

west coast and from Mexico in the south to North Dakota in the north (Fig 2c). Interestingly, the range of native spongy moths from Asia and Europe were larger than those of introduced spongy moths in North America. Additionally, the range of native spongy moths from Asia was approximately 1.22 times larger than those from Europe.

### 3.4. Range shifts of spongy moths

We did not observe any range expansions of the introduced spongy moths in North America compared with the native ones from Asia. The shared stabilizing ranges between introduced spongy moths in North America and the native ones from Asia covered an area of 112.16 × 10⁴ km² and were primarily projected within the range of the introduced spongy moths in North America (Fig 3a). The unfilling ranges of native spongy moths from Asia, relative to the introduced counterparts in North America, covered 1202.68 × 10⁴ km². They were primarily detected in Mexico as well as the southern part of Canada and the United States, except for the range of the introduced spongy moths in North America (Fig 3a). The *RR* between native spongy moths from Asia and the introduced ones in North America was about 11.72, indicating that the potential ranges of the former being approximately 11.72 times larger. The *IRS* between native spongy moths from Asia and the introduced ones in North America was 0.079, suggesting that the native spongy moths from Asia and the introduced ones in North America potentially occupied different range positions.

The areas of range expansions of the introduced spongy moths in North America compared with the native counterparts from Europe were mainly projected in Michigan, covering only about 0.15 × 10⁴ km² (Fig 3b). The shared stabilizing range between native spongy moths in the Europe and those introduced counterparts in North America was primarily projected within the range of the introduced spongy moths in North America, covering an area of 112.00 × 10⁴ km² (Fig 3b).

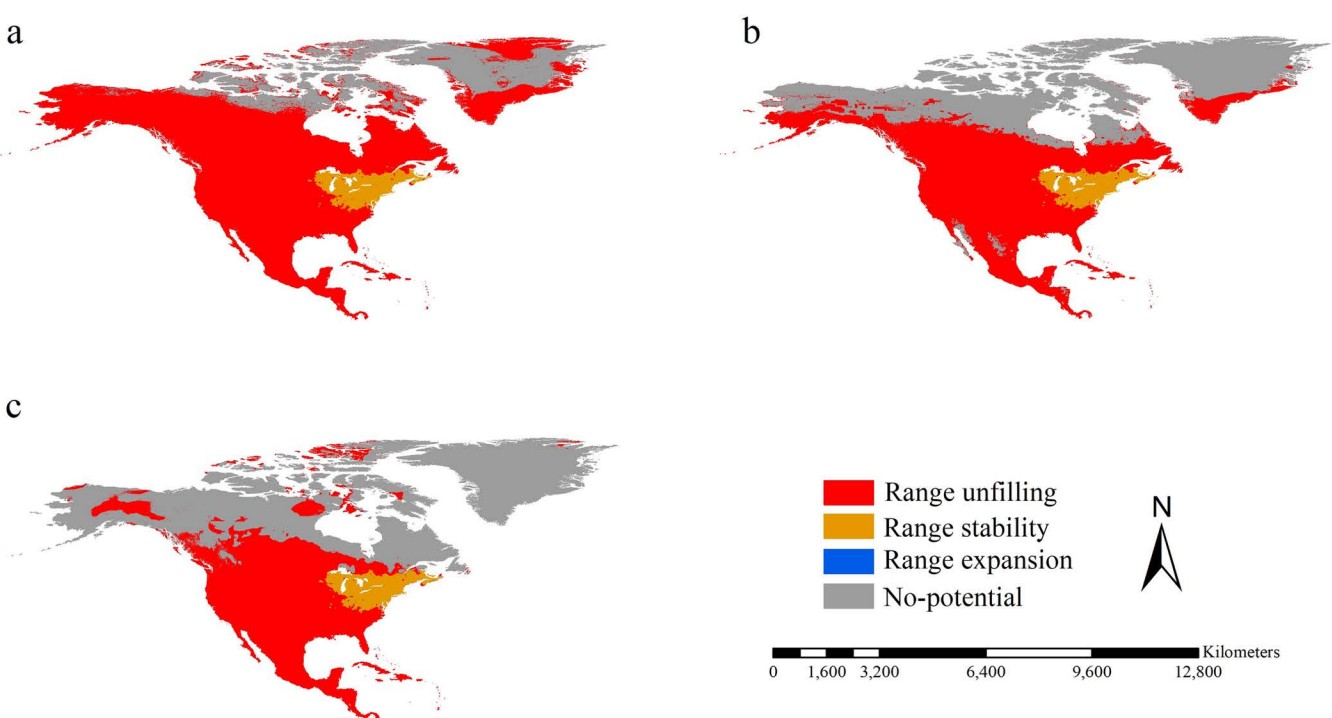

**Fig 3. Range shift of spongy moths on the spatial scale of North America.** Range shift between introduced spongy moths in North America and their native counterparts from Asia (a), and Europe (b). Red, orange, blue, and gray colors represent range unfilling, range stability, range expansion, and non-potential ranges, respectively. Reprinted from Worldclim under a CC BY 4.0 license, with permission from Worldclim, original copyright 2020-2024.

The unfilling range of native spongy moths from Europe relative to the introduced ones in North America covered 962.52 $\times 10^4$ km$^2$, and their patterns were similar to those of the unfilling range of the native ones from Asia relative to the introduced counterparts in North America (Fig 3b). The *RR* between native spongy moths from Europe and introduced spongy moths in North America was about 9.58, indicating that the potential ranges of the former being approximately 9.58 times larger. The *IRS* between introduced spongy moths in North America and their native counterparts in the Europe showed 0.094, suggesting that the introduced spongy moths in North America and the native counterparts in the Europe potentially had different range positions.

## 4. Discussion

In this study, the range of introduced spongy moths in North America was predominantly located in the central eastern region of the United States, including Maine, Michigan, New Brunswick, New Jersey, and New York, covering an area of 112.16 $\times 10^4$ km$^2$. However, compared with our projection, Morin et al. detected larger potential ranges for introduced spongy moths in North America [66]. They predicted the potential ranges based on interpolation of host species abundance. In contrast, our study employed SDMs established based on the niche theory, using occurrence records and seven different algorithms. Therefore, these differences could be attributed to the distinct methods employed, with the predicted potential ranges varying accordingly.

We observed that the introduced spongy moths in North America had smaller potential ranges than their native counterparts from Asia and Europe. This observation might be attributed to several factors. First, our observations supported this, as we noted that native spongy moths from Asia and Europe inhabited a wider latitudinal range, i.e., approximately 7.10°N - 64.44°N and 35.14°N - 62.00°N, respectively, compared with introduced spongy moths in North America (approximately 32.56°N - 52.00°N) (Fig 1). This heightened adaptability was reflected in our SDMs. Second, the introduced spongy moths in North America were identified as invasive pests and subjected to strict control measures upon their arrival [15,67], and had only one century of introduction history [1]. Additionally, the spongy moths in North American had experienced strict bottleneck that would substantially reduce plasticity level of its population [68]. Both of these two observations, to a certain extent, could lead to a decrease in their occurrences in North America and a subsequent reduction in their potential ranges.

Additionally, the potential ranges in North America for the spongy moths from Asia were 1.22 times larger than those from Europe. This difference might be linked to variations in their climatic adaptability. Notably, our occurrence datasets indicated that native spongy moths from Asia inhabited vast regions, spanning from South Asia to North Asia, covering a broad latitudinal range (57.34°). In contrast, the native spongy moths from Europe were observed across Europe, but covered smaller latitudinal spans (26.86°) (Fig 1). Further, the native spongy moths from Asia exhibited greater adaptability to diverse climates than their European counterparts (Fig 4).

Our study indicated that, among all the factors considered, climatic variables exerted the most significant effect on the range of spongy moths. This underscores the importance of monitoring the potential impact of future climatic changes on these spongy moths. The debate over the relative influence of climatic conditions and land-use on the range of invasive species continues [45]. On a continental scale, our study suggested that climatic variables had a substantial role in determining spongy moth range compared with the land use factors. While land use can modify host availability for spongy moths, its impact may be overshadowed by climatic factor. Relative influences of climatic and land use factors varied with spatial scales [46]. To some extent, our continental-scale study supported this argument. Examining the relative roles of climatic and land use factors in the range of this spongy moth was a novel contribution of our study. However, the general applicability of this pattern warrants further investigation in the future.

Topographical factors may serve as barriers to spongy moth dispersal [69]. However, this study indicated that topographical factors had lower importance values than climatic factors, implying weaker impacts on spongy moth ranges. This suggested that the influence of climatic adaptation superseded that of topographical factors. Alternatively, it could

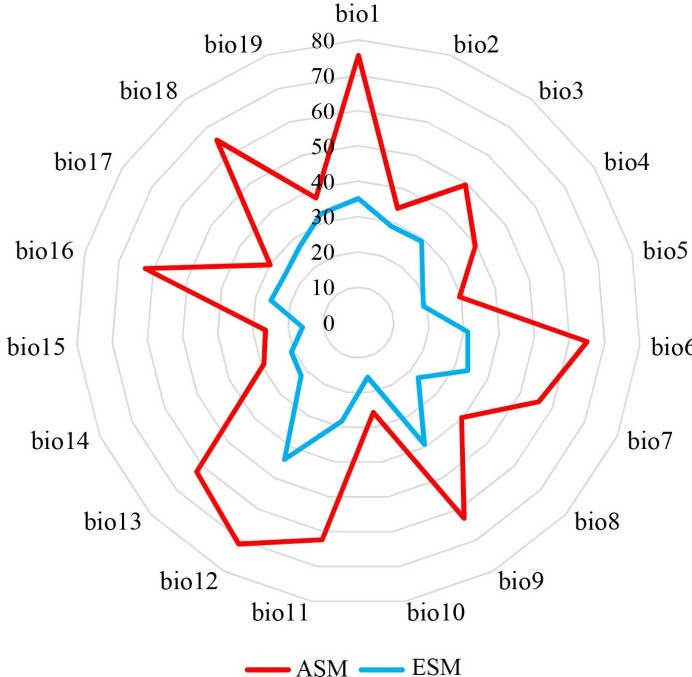

**Fig 4. Comparison of predictor ranges for spongy moths from Asia and Europe, with significant differences (*P* = 0.01).** Red and blue colors indicate the ranges of spongy moths from Asia and Europe, respectively.

also be attributed to well developed transportation networks that facilitated invasion, allowing this invasive pest to overcome the barrier effects of complex topographical patterns. Further, the risk of invasion by this pest should not be underestimated, even in regions with intricate topography.

As mentioned earlier, studies on the range shifts of invasive alien species are essential for effective invasion control. The areas in North America that native spongy moths could potentially inhabit, yet are unoccupied by the introduced spongy moths, were mainly projected to encompass most of the continent. This projection excludes regions favorable to the introduced spongy moths in North America and high latitude areas such as Nunavut and the central region of Greenland. This suggested that spongy moths may expand into these vast regions in the future. Therefore, significant attention is warranted to mitigate this risk, despite the current rarity of spongy moth occurrences in these areas.

While numerous studies have projected the distribution patterns of spongy moths under current and future scenarios [3,4,27,29,66,70,71], most have focused on the spatial patterns of habitat suitability for this invasive pest, offering valuable insights into invasion control in North America. By contrast, we established models to investigate potential range shifts between introduced North American spongy moths and native spongy moths from Asia and Europe, especially within the spatial context of North America. This study provided crucial insights into evaluating the invasion potential and risk associated with this invasive pest in North America. Additionally, it identified regions where spongy moths might potentially inhibit in North America but could not. Therefore, attention should be directed toward these priority regions to ensure the effective implementation of invasion control strategies against spongy moths in North America.

Finally, we should note that we projected the potential ranges of spongy moths on the basis of their occurrence records, which might not necessarily reflect their whole life history but just indicate that they occurred at some time and somewhere. For example, we found that approximately 90% of the occurrence records were observed or recorded between May and August. Also, although we projected the spongy moths' potential ranges, it did not necessarily mean that they

would complete their life cycle, such as their diapause, in their potential ranges. In other words, they could not continuously and fully occupy their potential ranges through their whole life history. Thus, their stable ranges might be substantially smaller than their potential ranges projected in the present study.

## 5. Conclusions

On the North American spatial scale, the potential ranges of native spongy moths were significantly larger from Asia and Europe than the current potential ranges of introduced spongy moths in North America, indicating a substantial invasive potential for this invasive pest in North America. This study also identified potential ranges that native spongy moths could inhabit in North America but are currently unoccupied. Overall, this study highlighted regions where spongy moths might expand into the future. Considering the greater influence of climatic variables on the ranges of spongy moths compared with land use and topographical factors, it is imperative to closely monitor the impacts of future climate changes on the ranges of these spongy moths. Notably, native spongy moths from Asia exhibited larger potential ranges than their native European counterparts. Hence, concerning range shifts, the native spongy moths from Asia might pose significant threats to forest ecosystems in North America than those from Europe.

## Supporting information

**S1 Table. Occurrences of the spongy moths.**
(XLSX)

**S2 Table. Importance scores of each predictor in the preliminary ecological niche models.**
(DOCX)

**S3 Table. Correlations among the thirty predictors of spongy moths in North America.**
(XLSX)

**S4 Table. Comparisons of AUC and TSS between real and null ecological niche models.**
(DOCX)

## Acknowledgments

**Disclaimer/Publisher's Note:** The statements, opinions and data contained in all publications are solely those of the individual author(s) and contributor(s) and not of MDPI and/or the editor(s). MDPI and/or the editor(s) disclaim responsibility for any injury to people or property resulting from any ideas, methods, instructions or products referred to in the content.

## Author contributions

**Conceptualization:** Jianmeng Feng.

**Data curation:** Yi Luo.

**Formal analysis:** Yi Luo, Changxi Li.

**Funding acquisition:** Xiaokang Hu.

**Investigation:** Yi Luo, Changxi Li.

**Methodology:** Yi Luo, Xiaokang Hu, Jianmeng Feng.

**Project administration:** Xiaokang Hu, Jianmeng Feng.

**Software:** Yi Luo, Changxi Li.

**Supervision:** Jianmeng Feng.

**Validation:** Yi Luo, Changxi Li.

**Visualization:** Yi Luo, Changxi Li.

**Writing – original draft:** Yi Luo, Xiaokang Hu, Jianmeng Feng.

**Writing – review & editing:** Yi Luo, Changxi Li, Xiaokang Hu, Jianmeng Feng.

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
