## [Decision Letter · Decision Letter 0]

28 Nov 2024

PONE-D-24-46649Spongy moths from Europe and Asia: Who could have higher invasion risk in North American? PLOS ONE

Dear Dr. Feng,

Thank you for submitting your manuscript to PLOS ONE. After careful consideration, we feel that it has merit but does not fully meet PLOS ONE’s publication criteria as it currently stands. Therefore, we invite you to submit a revised version of the manuscript that addresses the points raised during the review process.

We look forward to receiving your revised manuscript.

Kind regards,

Fang Zhu, Ph.D.

Academic Editor

PLOS ONE

Journal Requirements:

 National Science Foundation of China (Grant ID: 42263012).  

5. We note that Figures 1, 2, and 3 in your submission contain [map/satellite] images which may be copyrighted. All PLOS content is published under the Creative Commons Attribution License (CC BY 4.0), which means that the manuscript, images, and Supporting Information files will be freely available online, and any third party is permitted to access, download, copy, distribute, and use these materials in any way, even commercially, with proper attribution. For these reasons, we cannot publish previously copyrighted maps or satellite images created using proprietary data, such as Google software (Google Maps, Street View, and Earth). For more information, see our copyright guidelines: http://journals.plos.org/plosone/s/licenses-and-copyright.

a. You may seek permission from the original copyright holder of Figures 1, 2, and 3 to publish the content specifically under the CC BY 4.0 license.  

Additional Editor Comments :

Dear Authors,

I have received feedback from two reviewers regarding your manuscript titled "Spongy moths from Europe and Asia: Who could have higher invasion risk in North American?." After careful consideration of their comments, I would like to inform you that the reviewers have recommended major revisions.

Reviewer 2 has suggested only minor revisions, while Reviewer 1 has raised concerns that require more substantial changes to strengthen the manuscript. Specifically, Reviewer 1 expressed that while your study is a good one, addressing an important issue of species invasion and focusing on species from the "Top 100 Dangerous Invasive Species" list, they believe the study would benefit from a deeper integration of current knowledge in systematics, genetics, and ecology of the studied species. By incorporating a more thorough understanding of these areas, the manuscript would present a more comprehensive discussion and better align with the current state of research in this field.

Both sets of feedback are valuable, and I encourage you to carefully revise the manuscript, addressing all the comments and suggestions from both reviewers. Please ensure that you provide a detailed response to each comment, explaining how you have incorporated their suggestions or justifying any changes you have chosen not to make.

Once these revisions are completed, please submit the revised manuscript along with your response letter at your earliest convenience.

I look forward to receiving your revised manuscript.

Best,

Fang (Rose) Zhu

Reviewers' comments:

Reviewer's Responses to Questions

**Comments to the Author**

1. Is the manuscript technically sound, and do the data support the conclusions?

Reviewer #1: Partly

Reviewer #2: Yes

2. Has the statistical analysis been performed appropriately and rigorously? 

Reviewer #1: I Don't Know

Reviewer #2: N/A

3. Have the authors made all data underlying the findings in their manuscript fully available?

Reviewer #1: Yes

Reviewer #2: No

4. Is the manuscript presented in an intelligible fashion and written in standard English?

Reviewer #1: Yes

Reviewer #2: Yes

5. Review Comments to the Author

Reviewer #1: Dear editor and authors, I saw this MS earlier submitted via another journal. My decision is same as previously, this MS could be published after revision according to recommendations provided below.

Submitted MS provide the data of comparing the constructed models of L. dispar expansion in North America in the case of different population origins (from Asia or from Europe). The results of this comparison are given with in MS. I think it is good study which touch actual problem of species invasion, especially using as the example the species belonging to top 100 dangerous invasive species in the word. I also think that this study is acceptable for the auditory of considering journal. My criticism is belong to put this study more deep in actual knowledge about systematics, genetics and ecology of studied species that would make more close of discussion perspectives to observing reality at the moment. I think after this improving this MS might be accepted by the journal. See deleted comments below.

L38-41 Please refer also to more modern studies like Picq et al., 2023 who study SM populations in worldwide format demonstrating that population structure is more difficult than thought earlier.

L 51-53. Sorry but I miss the logic of this part of text. L.dispar was invade in north America from Western Europe (i.e. by nonflight females) that was confirmed by genetic studies (Wu et al., 2015, Molecular ecology, Picq et al., 2023 Evolutionary application). Does author discuss here the potential of Asian strain invasion or recorded fact of Asian spongy moth invasion in last sentence? If the last, please justify.

L 54-57. Yes, this is logical prediction but it could miss with facts. For example, both strains have the features of ballooning of first instar larvae (Martemyanov et al., 2019, Plos ONE), that could be more important for spreading than modest flight ability of females which are nonfeeding and can not flight for a long time (I,e, 750m/day, 5-6 days maximum, i.e. the distance of migration is 4-5km per season). If you will compare the moving of border range in North America (studies by Sharov or Tobin) or in Asia (Ponomarev et al., 2023, Insects for example), in both cases the distance will overcome the maximal flight ability of females. So flight ability of Asian population is less likely critically important for the speed of expansion in continental scale. Please consider this augments in more detail.

L75-82 When you discuss the effect of climate on L. dispar please put it in the context of microclimate. It is very important for such phenomenally adaptive species as LD and behavioral adaptation is one of the most important mechanism for such adaptation. For example formally supercooling point of L. dispar overwintered embryos is between -23-29C (Fält-Nardmann et al., 2018), however owing to laying the eggs masses under the snow cover (Ananko, Kolosov, 2021, Journal of Thermal Biology) or on the rocks in mounting areas (Ananko et al., 2022, Insects) that allow embryos avoid such temperature even outside temperature is significantly lower (-30C and lower). So overwinter temperature is unlikely to be an direct important climatic factor. I would say that microclimate condition are more limiting for LD expansion then general climate conditions. Please include this facts in the discussion of the theory.

L109-114 I think authors need to present this paragraph in more detail and more clear. Did I correct understand that you plan to use for created SDM the parameters from European and Asian populations of SM? This is important paragraph which will make more clear the following text.

L243 please decode MSS

Fig 2 please make the figure caption more informative and clear: which territory, which populations. Otherwise it sound confusing: c) “Range of native spongy moths in Europe” while figure present North American continent.

L 303-308 I agree that European SM and Asian SM possess wider range than American population but I think that for Asian SM the border is too wide and at south border (at least according to figure 1) include closely related species such as Lymantria obfuscata (India), L. xylina (China), L. albescens (Japan). Please check it.

L 311-312 you could also argue that North American population went though strict bottleneck (Wu et al., 2015) that would decrease plasticity level of invader population,

L 313-314. I am not native English speaking researcher but I think to avoid confusing between insects sources and range (territory) better to write “spongy moth FROM Asia/Europe” if I correct understand what authors want to say. If it is correct please use this correction for other parts of text whenever it was used.

L 313-320. I hope this data based on true L. dispar records (i.e. not on closely related species) with correct species identification. This question address to the data sources used for this study rather than to authors of this study. To me the south border of Asian SM range includes some closely related species. In turn, I did not find north Africa locations for European spongy moth populations (see Picq et al., 2023) what will make the latitude gradient equal for both strains Asian and European subspecies.

L 322-327 I miss what authors would say here. You could use more informative reference (Djoumad et al., Scientific report) describing unique mt genome of Caucasus populations of Lymatria dispar which possibly will described as separate subspecies in future (see Picq et al., 2023). But anyway I do not understand what kind of comparison here authors would make. I confuse in the context of following text and miss what authors want to highlight. Please make it more clear and logic.

L 332-337. Again, maximal female flight ability of Asian strain measured in Km per season is significantly low to compare with speed of spreading of range border. For example see recent study of Ponomarev et al., 2023, Insects where speed of border range moving is 50 km/year and studied population belong to European spongy moth according to genotyping by sequencing markers (Picq et al., 2023). So female flight ability is mostly local tactics of the species rather than spreading strategy. Typical example is population in Altai mountings which formally belong to European genotypes, but the life cycle is obligatory related with female flight ability allowing to lay eggs on the rocks remote from host plant (Ananko at al., 2022, Insects). So this traditional features of Asian SM populations is less likely to seriously invest to spreading of population at continental scale.

L380-389 As I read it I think that logic is broken here. If most data used in the analysis were collected in the part of the season this mean that stable range could be less then described according to the reasons described in this paragraph. Possibly I misunderstand the message of author, anyway this mean that rewording is needed.

I think after major revision of this MS it could be acceptable for the journal.

Reviewer #2: Major Comments:

Population Structure:

What is the population structure of spongy moths in North America? Does it represent a combination of European and Asian populations, or does it constitute a distinct, new population?

Methods:

The description of the methods is unclear. Please provide additional details, specifically clarifying the purpose of each method and how it was implemented.

Figures:

For Figure 2, there are 4 sub-figures, but the authors only explain 3. Similarly, Figure 3 contains 3 sub-figures, yet the authors describe only 2. Please ensure all sub-figures are adequately addressed in the text.

Minor Comments:

Typographical Errors:

Typos in lines 37, 43, 48, and 56 et al, need correction.

Line 111:

What does "SDMs" stand for? Please define the term for clarity.

Line 157:

How were the importance scores of the remaining predictors calculated in the final species distribution models and the preliminary ecological niche models? Please elaborate.

6. PLOS authors have the option to publish the peer review history of their article (what does this mean? ). If published, this will include your full peer review and any attached files.

**Do you want your identity to be public for this peer review?** For information about this choice, including consent withdrawal, please see our Privacy Policy .

Reviewer #1: **Yes: ** Martemyanov Vyacheslav

Reviewer #2: No

---

## [Author Response · Author response to Decision Letter 1]

27 Dec 2024

Responses to Reviewer #1

Dear reviewer:

We sincerely appreciate your insightful comments and all the suggestions you provided for our manuscript, which have highlighted several shortcomings in our manuscript. Additionally, your profound comments on our topic demonstrate your expertise in the field, which we greatly admire. In our revised version, we have addressed all of your concerns. We have also provided detailed responses to the concerns you raised, which you can find in the following section of my point-by-point responses. Finally, we would like to express our gratitude for your diligent efforts in reviewing our manuscript. We hope that our revisions meet your expectations for the manuscript. Of course, if you have any further valuable suggestions, we sincerely welcome them, and we will make additional revisions based on your feedback.

Reviewer #1: Dear editor and authors, I saw this MS earlier submitted via another journal. My decision is same as previously, this MS could be published after revision according to recommendations provided below.

Submitted MS provide the data of comparing the constructed models of L. dispar expansion in North America in the case of different population origins (from Asia or from Europe). The results of this comparison are given with in MS. I think it is good study which touch actual problem of species invasion, especially using as the example the species belonging to top 100 dangerous invasive species in the word. I also think that this study is acceptable for the auditory of considering journal. My criticism is belong to put this study more deep in actual knowledge about systematics, genetics and ecology of studied species that would make more close of discussion perspectives to observing reality at the moment. I think after this improving this MS might be accepted by the journal. See deleted comments below.

Our responses: Thanks for your very professional comments. We have tried our best to revised our MS. If you have any further valuable suggestions, we sincerely welcome them, and we will make additional revisions.

L38-41 Please refer also to more modern studies like Picq et al., 2023 who study SM populations in worldwide format demonstrating that population structure is more difficult than thought earlier.

Our responses: Thanks for your very professional comments. We have added more recent studies to our revised MS at line of 41, including those by Picq et al. (2023). Thanks!

L 51-53. Sorry but I miss the logic of this part of text. L.dispar was invade in north America from Western Europe (i.e. by nonflight females) that was confirmed by genetic studies (Wu et al., 2015, Molecular ecology, Picq et al., 2023 Evolutionary application). Does author discuss here the potential of Asian strain invasion or recorded fact of Asian spongy moth invasion in last sentence? If the last, please justify.

Our responses: Thanks for your very constructive comments. We wrote this part of text for comparing the different flight ability of these two strains. However, with a reference to your following comment, it is not a key or suitable point. Therefore, we have removed it, and compared their invasion history and host ranges, which might induce their different invasion potentials at lines of 46-52.

L 54-57. Yes, this is logical prediction but it could miss with facts. For example, both strains have the features of ballooning of first instar larvae (Martemyanov et al., 2019, Plos ONE), that could be more important for spreading than modest flight ability of females which are nonfeeding and can not flight for a long time (I,e, 750m/day, 5-6 days maximum, i.e. the distance of migration is 4-5km per season). If you will compare the moving of border range in North America (studies by Sharov or Tobin) or in Asia (Ponomarev et al., 2023, Insects for example), in both cases the distance will overcome the maximal flight ability of females. So flight ability of Asian population is less likely critically important for the speed of expansion in continental scale. Please consider this augments in more detail.

Our responses: Thanks for your very constructive comments. We fully accept your professional comment, and removed this phrase. In our revisions, we compared their invasion history and host ranges which might induce their different invasion potentials in North America at lines of 46-52.

L75-82 When you discuss the effect of climate on L. dispar please put it in the context of microclimate. It is very important for such phenomenally adaptive species as LD and behavioral adaptation is one of the most important mechanism for such adaptation. For example formally supercooling point of L. dispar overwintered embryos is between -23-29C (Fält-Nardmann et al., 2018), however owing to laying the eggs masses under the snow cover (Ananko, Kolosov, 2021, Journal of Thermal Biology) or on the rocks in mounting areas (Ananko et al., 2022, Insects) that allow embryos avoid such temperature even outside temperature is significantly lower (-30C and lower). So overwinter temperature is unlikely to be an direct important climatic factor. I would say that microclimate condition are more limiting for LD expansion then general climate conditions. Please include this facts in the discussion of the theory.

Our responses: Thanks for your very constructive comments. We fully accept your professional comments. We have made revisions by saying that microclimate conditions are more limiting for LD expansion than general climate conditions at lines 73-82. Thanks!

L109-114 I think authors need to present this paragraph in more detail and more clear. Did I correct understand that you plan to use for created SDM the parameters from European and Asian populations of SM? This is important paragraph which will make more clear the following text.

Our responses: Thanks for your professional comments. We agree with you. We have rewritten this paragraph in more detail. In our study, we employed species distribution model tools to examine the potential range of the introduced spongy moths in North America. We also built SDMs for the native spongy moths in Asia and Europe, separately, and the models were generally determined by their traits or adaptation to the environmental conditions. Then, we transferred the models to North America, and projecting their potential ranges there, and the revisions could be found at lines of 117-127. Thanks!

L243 please decode MSS

Our responses: Thanks for your kind reminding. We have decoded MSS at line of 200.

Fig 2 please make the figure caption more informative and clear: which territory, which populations. Otherwise it sound confusing: c) “Range of native spongy moths in Europe” while figure present North American continent.

Our responses: Thanks for your very kind reminding. We have made revisions on Figure 2. Specifically, (a) Potential ranges of introduced spongy moths in North America. (b) Projected potential ranges in North America for the spongy moths from Asia. (c) Projected potential ranges in North America for the spongy moths from Europe. Thanks!

L 303-308 I agree that European SM and Asian SM possess wider range than American population but I think that for Asian SM the border is too wide and at south border (at least according to figure 1) include closely related species such as Lymantria obfuscata (India), L. xylina (China), L. albescens (Japan). Please check it.

Our responses: Thanks for your very rigorous comments and kind reminding, which push us to review and check our source occurrence data. From GBIF, we found that the southmost occurrence in Asia was in Sri Lanka (7.1N, 80.1E), from which the occurrence of our target species was detected on 10th Aug 2021 (https://www.gbif.org/occurrence/3466823587). It, to certain extent, was supported by a study (https://doi.org/10.1038/s41598-019-57020-7) which argued that the suitability for this species was considered high in southern parts of India and Sri Lanka. We fully understand your professional inquiry. Maybe, a closely related species was mistakenly identified as Lymantria dispar Linnaeus. Also, maybe it truly is Lymantria dispar Linnaeus as a result of one of accidental events in the spreading of this species. However, the reliability of this occurrence record depends on the professional ability of the observer, and we have not solid evidence to definitely and easily deny its reliability. However, if you tend to remove it, we will delete it in our future revisions. Thanks again.

L 311-312 you could also argue that North American population went though strict bottleneck (Wu et al., 2015) that would decrease plasticity level of invader population,

Our responses: Thanks for your very professional comments. We fully agree with you, and have made revisions at lines of 341-344.

L 313-314. I am not native English speaking researcher but I think to avoid confusing between insects sources and range (territory) better to write “spongy moth FROM Asia/Europe” if I correct understand what authors want to say. If it is correct please use this correction for other parts of text whenever it was used.

Our responses: Thanks for your very kind reminding. Your comment clearly reflects what we want to say, and we fully agree with you. Accordingly, we have made relevant revisions through our MS, making it clearer and smoother. Thanks again!

L 313-320. I hope this data based on true L. dispar records (i.e. not on closely related species) with correct species identification. This question address to the data sources used for this study rather than to authors of this study. To me the south border of Asian SM range includes some closely related species. In turn, I did not find north Africa locations for European spongy moth populations (see Picq et al., 2023) what will make the latitude gradient equal for both strains Asian and European subspecies.

Our responses: Thanks for your very professional comments, and we fully agree with and understand your professional inquiry. The southmost occurrence of our target species in Asia was recorded in Sri Lanka (7.1N, 80.1E, 10th Aug 2021, www.gbif.org/occurrence/3466823587), which, to certain extent, was supported by a study (https://doi.org/10.1038/s41598-019-57020-7) arguing that the suitability for this species was considered high in southern parts of India and Sri Lanka. However, we have to acknowledge that maybe, a closely related species was mistakenly identified as Lymantria dispar Linnaeus. Also, maybe it truly is Lymantria dispar Linnaeus as a result of one of accidental events in the spreading of this species. However, the reliability of this occurrence record depends on the professional ability of the observer, and we have not solid evidence to definitely and easily deny its reliability. However, if you tend to remove it, we will do it in our next revision version. Moreover, in our study, we included three populations of Lymantria dispar Linnaeus as our study objects, i.e., populations of Lymantria dispar from Asia, Europe and North America, and populations of Lymantria dispar from Asia and Europe could not necessarily equal to the Asian and European spongy moths, respectively. Therefore, in our study, European spongy moth populations North Africa was not incorporated into our study. We hope our explanation could satisfy you. If you have further inquiries on it, please don’t hesitate to tell us in future revisions. We appreciate your professional comments, and have revised our phrases to make it clearer.

L 322-327 I miss what authors would say here. You could use more informative reference (Djoumad et al., Scientific report) describing unique mt genome of Caucasus populations of Lymatria dispar which possibly will described as separate subspecies in future (see Picq et al., 2023). But anyway I do not understand what kind of comparison here authors would make. I confuse in the context of following text and miss what authors want to highlight. Please make it more clear and logic.

Our responses: Thanks for your very professional comments which make us reconsider the contribution of this paragraph to our topic. We agree with your comment that this paragraph seems irrelevant to our topic. Therefore, we have removed it from our revised MS. Thanks!

L 332-337. Again, maximal female flight ability of Asian strain measured in Km per season is significantly low to compare with speed of spreading of range border. For example see recent study of Ponomarev et al., 2023, Insects where speed of border range moving is 50 km/year and studied population belong to European spongy moth according to genotyping by sequencing markers (Picq et al., 2023). So female flight ability is mostly local tactics of the species rather than spreading strategy. Typical example is population in Altai mountings which formally belong to European genotypes, but the life cycle is obligatory related with female flight ability allowing to lay eggs on the rocks remote from host plant (Ananko at al., 2022, Insects). So this traditional features of Asian SM populations is less likely to seriously invest to spreading of population at continental scale.

Our responses: Thanks for your very professional comments, and we agree with you. Owing to your valuable comment, we realized that using female flight ability to explain the difference of range sizes between the populations of Lymantria dispar from Asia and Europe is misleading. Therefore, we have removed it from our revised MS. Thanks!

L380-389 As I read it I think that logic is broken here. If most data used in the analysis were collected in the part of the season this mean that stable range could be less then described according to the reasons described in this paragraph. Possibly I misunderstand the message of author, anyway this mean that rewording is needed.

Our responses: Thanks for your very professional comments, and we fully agree with you and have made revisions at lines of 420-424.

I think after major revision of this MS it could be acceptable for the journal.

Responses to Reviewer #2

Dear Reviewer:

We greatly appreciate your positive feedback on our manuscript, which is very exciting for us. Additionally, we are grateful for the valuable comments and suggestions you provided for our manuscript. Following your suggestions, we have carefully revised the content of the manuscript and thoroughly addressed your queries on certain sections. Please refer to the point-by-point responses for specific details.

Lastly, we sincerely thank you for your diligent efforts in reviewing our manuscript. We hope that the revisions made to the manuscript meet your expectations. However, if you have any further suggestions or valuable insights, we warmly welcome them. We will use your feedback as a basis for any necessary adjustments and improvements.

Reviewer #2: Major Comments:

Population Structure:

What is the population structure of spongy moths in North America? Does it represent a combination of European and Asian populations, or does it constitute a distinct, new population?

Our responses: Thanks for your very professional comments. According to the arguments by some scientists, population of spongy moths in North America is a combination of European and Asian populations. We have made revisions at lines of 130-131. Thanks!

Methods:

The description of the methods is unclear. Please provide additional details, specifically clarifying the purpose of each method and how it was implemented.

Our responses: Thanks for your very constructive comments, and we fully agree with you. We have added more detailed description for the methods.

Figures:

For Figure 2, there are 4 sub-figures, but the authors only explain 3. Similarly, Figure 3 contains 3 sub-figures, yet the authors describe only 2. Please ensure all sub-figures are adequately addressed in the text.

Our responses: Thanks for your very kind reminding. These are our mistakes. In Figure 2, 2-a was an unnecessary or redundant sub-figure, and in Figure 3 3-a was also an unnecessary or redundant sub-figure. They were subfigures in our previous versions of MS, representing the potential ranges of the spongy moths from

---

## [Decision Letter · Decision Letter 1]

21 Feb 2025

Spongy moths from Europe and Asia: Who could have higher invasion risk in North American?

PONE-D-24-46649R1

Dear Dr. Feng,

We’re pleased to inform you that your manuscript has been judged scientifically suitable for publication and will be formally accepted for publication once it meets all outstanding technical requirements.

Kind regards,

Fang Zhu, Ph.D.

Academic Editor

PLOS ONE

Additional Editor Comments (optional):

Authors Feng et al. have done excellent job addressing the reviewers' comments and revising the manuscript. I have no further comments, and I believe the manuscript is now suitable for publication.

Reviewers' comments:

Reviewer's Responses to Questions

**Comments to the Author**

1. If the authors have adequately addressed your comments raised in a previous round of review and you feel that this manuscript is now acceptable for publication, you may indicate that here to bypass the “Comments to the Author” section, enter your conflict of interest statement in the “Confidential to Editor” section, and submit your "Accept" recommendation.

Reviewer #1: All comments have been addressed

Reviewer #3: All comments have been addressed

2. Is the manuscript technically sound, and do the data support the conclusions?

Reviewer #1: Yes

Reviewer #3: Yes

3. Has the statistical analysis been performed appropriately and rigorously? 

Reviewer #1: Yes

Reviewer #3: Yes

4. Have the authors made all data underlying the findings in their manuscript fully available?

Reviewer #1: Yes

Reviewer #3: Yes

5. Is the manuscript presented in an intelligible fashion and written in standard English?

Reviewer #1: Yes

Reviewer #3: Yes

6. Review Comments to the Author

Reviewer #1: Authors address all comments. Authors are right that the accuracy of species identifying is independent from their efforts so range scale of L. dispar is based on the data referred one the certain studies. Thus to me current version of MS much more clarified and is acceptable for the publication in PLoS ONE.

Reviewer #3: (No Response)

7. PLOS authors have the option to publish the peer review history of their article (what does this mean? ). If published, this will include your full peer review and any attached files.

**Do you want your identity to be public for this peer review?** For information about this choice, including consent withdrawal, please see our Privacy Policy .

Reviewer #1: No

Reviewer #3: No

---

## [Editor Report · Acceptance letter]

PONE-D-24-46649R1

PLOS ONE

Dear Dr. Feng,

I'm pleased to inform you that your manuscript has been deemed suitable for publication in PLOS ONE. Congratulations! Your manuscript is now being handed over to our production team.

Kind regards,

on behalf of

Dr. Fang Zhu

Academic Editor

PLOS ONE